# Autophagy in α-Synucleinopathies—An Overstrained System

**DOI:** 10.3390/cells10113143

**Published:** 2021-11-12

**Authors:** Lisa Fellner, Elisa Gabassi, Johannes Haybaeck, Frank Edenhofer

**Affiliations:** 1Department of Genomics, Stem Cell Biology and Regenerative Medicine, Institute of Molecular Biology & CMBI, Leopold-Franzens-University Innsbruck, 6020 Innsbruck, Austria; elisa.gabassi@uibk.ac.at (E.G.); frank.edenhofer@uibk.ac.at (F.E.); 2Institute of Pathology, Neuropathology and Molecular Pathology, Medical University of Innsbruck, 6020 Innsbruck, Austria; johannes.haybaeck@i-med.ac.at; 3Diagnostic & Research Center for Molecular Biomedicine, Institute of Pathology, Medical University of Graz, 8036 Graz, Austria

**Keywords:** alpha-synuclein, Parkinson’s disease, multiple system atrophy, autophagy, neurons, oligodendroglia

## Abstract

Alpha-synucleinopathies comprise progressive neurodegenerative diseases, including Parkinson’s disease (PD), dementia with Lewy bodies (DLB), and multiple system atrophy (MSA). They all exhibit the same pathological hallmark, which is the formation of α-synuclein positive deposits in neuronal or glial cells. The aggregation of α-synuclein in the cell body of neurons, giving rise to the so-called Lewy bodies (LBs), is the major characteristic for PD and DLB, whereas the accumulation of α-synuclein in oligodendroglial cells, so-called glial cytoplasmic inclusions (GCIs), is the hallmark for MSA. The mechanisms involved in the intracytoplasmic inclusion formation in neuronal and oligodendroglial cells are not fully understood to date. A possible mechanism could be an impaired autophagic machinery that cannot cope with the high intracellular amount of α-synuclein. In fact, different studies showed that reduced autophagy is involved in α-synuclein aggregation. Furthermore, altered levels of different autophagy markers were reported in PD, DLB, and MSA brains. To date, the trigger point in disease initiation is not entirely clear; that is, whether autophagy dysfunction alone suffices to increase α-synuclein or whether α-synuclein is the pathogenic driver. In the current review, we discuss the involvement of defective autophagy machinery in the formation of α-synuclein aggregates, propagation of α-synuclein, and the resulting neurodegenerative processes in α-synucleinopathies.

## 1. Introduction

The term α-synucleinopathies (ASPs) was introduced to categorize a group of disorders that feature pathological α-synuclein (α-syn) accumulations in neuronal and/or glial cells throughout the central nervous system (CNS). Inclusions of α-syn and neuroinflammatory response have also been reported to occur in the peripheral nervous system (PNS) [1,2,3]. ASPs cover Parkinson’s disease (PD), dementia with Lewy bodies (DLB), and multiple system atrophy (MSA) and are adult-onset, progressive, neurodegenerative diseases [4]. They differ in the occurrence of α-syn-positive deposits; in PD and DLB, the abnormal aggregates appear preferentially in neurons—termed Lewy bodies (LBs) and Lewy neurites [5,6], whereas MSA displays the α-syn-positive aggregates predominantly in oligodendroglial cells, which are called glial cytoplasmic inclusions (GCIs) [7,8]. The accumulation is thought to be caused by a diminished degradation of the aberrant form of α-syn in neurons and glia, leading to increased levels of toxic α-syn species, inducing neurodegenerative events [9,10,11]. There is accumulating evidence that neurodegeneration can also be initiated and caused by inflammatory events. Microglial and astroglial cells can be activated by various α-syn forms, among others (e.g., brain injury and infections), leading to an immunological reaction in the brain. If the insult lasts too long, as in neurodegenerative diseases, a kind of feedback loop evolves and microglia and astroglia can become over-activated, so-called reactive micro- and astrogliosis, and lead to neuronal loss as well [12,13,14].

α-syn is a neuronal protein and is localized predominantly in presynaptic terminals of the striatum, hippocampus, thalamus, cerebellum, and neocortex, respectively [15,16]. Its function in a healthy brain is not fully elucidated to date, but includes regulating synaptic vesicle maintenance and recycling, as well as a role in synaptic plasticity [17,18,19]. The protein is part of a distinctive family of genes that includes SNCA—α-syn, SNCB—β-syn, and SCNG—γ-syn [20,21]. α-syn contains 140 amino acids and comprises three distinctive regions: (1) residues 1–60 form an amphipathic N-terminus including repeats of the conserved KTKEGV motif, (2) residues 61–95 assemble to form a non-amyloid β-component (NAC) region—a hydrophobic area, and (3) residues 96–140 build an acidic C-terminus [22]. The NAC region (residues 71–82) and a short motif in the N-terminal region (residues 36–42) in the center of the α-syn gene were shown to be involved in its aggregation [23,24]. The structure of α-syn includes a motif for binding of the heat shock cognate 70 (Hsc70) chaperone, VKKDQ (KFERQ-like, residues 95–99), leading to the degradation of α-syn via the chaperone-mediated autophagy (CMA) pathway (Figure 1) [25,26,27].

Pathways contributing to the degradation of wild type or soluble α-syn in healthy cells include the autophagic machinery and the proteasomal degradation pathway [28,29], whereby the autophagy pathway was recognized as the more common degradation process of misfolded α-syn [30,31]. Regarding the pathogenesis of ASPs, a contribution of a disrupted autophagy-lysosomal pathway (ALP) has been demonstrated to play a major role in the formation of α-syn-positive aggregations [32,33,34,35]. The autophagic system in these neurodegenerative disorders may reach a saturation point in which protein aggregates are not degraded anymore. In a vicious cycle, protein aggregates may be also responsible for the sequestration of autophagy proteins that can no longer contribute to the formation of the autophagic machinery [11].

The ALP in the cell is an adaptive machinery responsible for the degradation of defective and redundant cellular components, as well as long-lived proteins, thereby providing the cell nutrients for indispensable cellular functions, including fasting and other forms of stress and, furthermore, cytoprotection. Autophagy is a cellular process that needs stringent control to secure the correct response to different cellular stimuli. The term autophagy comprises three different forms: (1) microautophagy, (2) macroautophagy, and (3) CMA. Microautophagy will not be further discussed in the current review, as no data exist on microautophagy involvement in the degradation of α-syn. In contrast, the other autophagy forms, i.e., macroautophagy and CMA, seem to be relevant for the α-syn degradation process [36].

Macroautophagy (commonly called autophagy) begins with the generation of autophagosomes, which are double-lipid membrane structures involved in degrading cytosolic constituents, such as organelles and proteins. In a next step, the autophagosomes fuse with lysosomes, yielding autophagolysosomes, in which the proteins or organelles are subsequently degraded by hydrolases. The developing free amino acids are then again available to the cell for the building of new cellular constituents. Numerous proteins take part in the formation of the autophagosome, including the 15 so-called autophagy-related proteins (ATGs, AuTophaGy), first described in yeast [37]. Among others, the autophagy-related protein microtubule-associated protein 1 light chain 3B (LC3B) and Beclin-1 are significant players in autophagosome generation and are common markers used in measuring macroautophagy activity. The lipidated form of LC3, LC3-II, is found specifically on the membrane of autophagosomes [38]. Furthermore, the ubiquitin-binding protein, p62, a receptor for recognizing poly-ubiquitinated proteins and LC3, act together and escort these protein aggregates to the autophagosomes [39,40]. Macroautophagy is controlled by different upstream proteins, including UNC-51-like kinases (ULK1, ULK2), vacuolar protein sorting-associated protein 34 (VPS34), and autophagy/beclin1 regulator 1 (AMBRA1). Macroautophagy commences by the activation or modification of these upstream proteins, thereby blocking the mammalian target of rapamycin (mTOR) [41]. For a detailed report on macroautophagy and its key players, see the reviews from Dikic I. and Elazar Z. and Yang Z. and Klionsky D.J. [42,43].

CMA is a highly selective autophagy pathway and, compared with micro- and macroautophagy, no vesicle formation occurs during the degradation process. Importantly, for the recognition of the substrate by a group of chaperones (including Hsc70 chaperone), a CMA targeting motif is necessary. Only proteins containing this motif, a sequence biochemically related to the KFERQ pentapeptide, will be recognized and escorted directly to the lysosome [44]. At the lysosomal membrane, interaction of the proteins with the cytosolic end of the lysosome-associated membrane protein type 2A (Lamp-2A) will occur. Lamp-2A is a transmembrane protein that helps to translocate the substrate through the lysosomal membrane, where a rapid degradation process by hydrolytic enzymes takes place [45].

As the accumulation of α-syn is one of the fundamental events in the initiation and advancement of PD, DLB, and MSA, unravelling the cause behind the aggregation process is highly desired to understand these neurodegenerative diseases and to generate new medication that will halt the worsening of ASPs. In the current review, we will highlight recent findings that claim an involvement of autophagy, including macroautophagy and CMA, in disease initiation and progression of ASPs. Thereby, we will elaborate on differences and similarities regarding autophagy impairment in neuronal ASPs, including PD and DLB, compared with oligodendroglial ASPs, namely MSA. Yet even though there are differences regarding the exact mechanisms behind α-syn aggregation, autophagy dysfunction seems to be involved in all three clinically distinct diseases, i.e., PD, DLB, and MSA.

## 2. Autophagic Dysregulation in Neuronal ASPs

PD is a common neurodegenerative disease, second after Alzheimer’s disease (AD) for incidence and affecting approximately 2–3% of the world’s population over the age of 65 [46,47]. PD symptoms include motor dysfunctions such as resting tremor, rigidity, and bradykinesia, as well as non-motor manifestations like cognitive impairment, mood, sleep disorders, and pain [48,49]. In particular, motor dysfunctions are the consequence of neuronal death in the substantia nigra (SN). Dopaminergic neuronal loss and accumulation of LB with fibrillar α-syn-aggregates represent the major well-established hallmarks of PD. Therapeutic approaches include treatments to alleviate symptoms or delay the worsening of the clinical phenotype, mainly based on long-term dopamine-based drugs like levodopa. However, to date, no treatment is reported to stop or reverse the progression of the disease [50].

DLB is mainly characterized by the presence of diffuse LBs in the SN, the cerebral cortex, and the subcortical nuclei, with a more marked cortex pathology compared with PD [51]. However, PD and DLB share many similarities in LBs’ occurrence and localization and can be distinguished mainly thanks to diagnostic criteria. In particular, clinical manifestations of DLB include cognitive deficits associated with a Parkinsonian phenotype, and fluctuations in attention and awareness, often accompanied by visual hallucinations [52,53]. The treatment of DLB combines the use of levodopa for Parkinsonism with cholinergic therapy to alleviate cognitive impairment [54], yet no treatments exist thus far that could halt the progression of DLB. As described in the introduction, autophagy is believed to be a major contributor regarding the clearance of aggregates of mutant proteins. However, the autophagic mechanisms involved in the pathogenesis and progression of DLB and PD remain ambiguous. To date, there is no study reporting that PD and DLB share the same pathogenic process; nonetheless, many common features have been observed in patients in the context of autophagy, as will be described below. While early reports investigated the involvement of the proteasome system in the degradation of α-syn aggregates [55], it is widely accepted nowadays that alterations in the autophagic pathways may be preferentially involved in neurodegenerative diseases, including PD [56].

### 2.1. Autophagy Involvement in Human Pathology of PD and DLB

The first indication that autophagy plays a role in neurodegeneration arose from several observations of accumulated autophagosomes in the brains of patients affected by diverse disorders, including PD, AD, and Huntington’s disease. Signs of autophagic degradation were detected in nigral neurons of PD patients; in particular, aggregated autophagosomes and aberrant lysosomes were found in post-mortem PD samples [56,57]. Furthermore, it was observed that, in LBs, α-syn colocalized with the autophagy marker LC3, and LC3-II levels were elevated in brain tissue from PD patients [58,59]. Along with the aforementioned markers, p62 also was detected in LBs from patients, further supporting the involvement of macroautophagy in the progression of PD [60]. Interestingly, in post-mortem brains from PD and DLB patients, it was observed that numerous upstream autophagosomal proteins, namely, ULK1, ULK2, Beclin-1, VPS34, and AMBRA1, are included in LBs and up-regulated in PD [61]. Another study with human brain samples showed that levels of the lysosomal type 5 P-type ATPase ATP13A2 were decreased in PD neurons, with ATP13A2 being sequestered in LBs [62]. ATP13A2 was reported to participate in transmembrane lysosomal transport and mutations affecting the *ATP13A2* gene are linked to autosomal recessive familial PD [62,63]. In a recent study, the authors showed, using super-resolution microscopy, the existence of a crowded environment of organelles in PD patients’ brains, highlighting the presence of mitochondria, lysosomes, and autophagosomes [64].

Besides, in the context of CMA, samples from SN of PD patients displayed a selective loss of CMA markers, namely HSC70 protein and Lamp-2A [59,65]. Interestingly, one study showed that Lamp-2A levels are reduced concomitantly with the increase in α-syn levels at early stages of PD, suggesting that impairments in the CMA machinery could occur even before the large accumulation of α-syn takes place [65]. It has been shown that monomers and dimers of α-syn are mainly degraded by CMA compared with the oligomeric form [31,66]. Therefore, when the CMA machinery is compromised, monomers and dimers of α-syn tend to further aggregate, giving rise to even more oligomers. This hypothesis is consistent with several independent observations reporting that impaired CMA results in α-syn aggregation. This is true, for example, in the presence of mutations affecting proteins such as leucine-rich repeat kinase 2 (LRRK2), ubiquitin C-terminal hydrolase L1 (UCHL1), and DJ-1 (PARK7), all of which are responsible for the onset of familial forms of PD and can be degraded by CMA [59,66,67,68]. One study showed that PD brains with mutant LRRK2 protein create a self-perpetuating inhibitory effect on CMA [69]. Additionally, the involvement of microRNAs (miRNAs) in the regulation of CMA in PD has also been reported. The analysis of PD patients’ brains showed that six miRNAs predicted to regulate Lamp-2A and HSC70 were increased in the SN of patients, suggesting that the miRNA-induced dysregulation of the autophagic machinery plays an important role in PD pathogenesis and may represent a novel therapeutic target [70].

Furthermore, both macroautophagy and CMA rely on the presence and function of lysosomal enzymes, which have been reported to be impaired in post-mortem brains of PD patients. In particular, patients with mutations affecting the glucocerebrosidase gene (GBA) present a decrease in the activity of the lysosomal β-glucocerebrosidase (GCase) enzyme in several brain areas [71]. Moreover, Chu Y. and colleagues reported that PD nigral neurons exhibit lower levels of the lysosome associated membrane protein 1 (LAMP1), the lysosomal hydrolase Cathepsin D (CatD), and heat shock protein 73 (HSP73) in comparison with control brains [72]. Interestingly, immunoreactivity of GCase along with α-mannosidase and β-mannosidase, other lysosomal enzymes, was diminished in PD patients’ cerebrospinal fluid (CSF) [73,74]. A reduction in other autophagy markers, such as LC3B, Lamp-2A, and Beclin-1, was also observed in the CSF from subjects with PD [75]. Finally, peripheral blood mononuclear cells (PBMCs) isolated from PD patients can also be used to detect the presence of biomarkers, exhibiting diminished levels of HSC70 and Lamp-2A along with an up-regulation of LC3, ULK1, Beclin-1, and AMBRA1 [76,77,78,79]. Interestingly, a recent study investigated the methylation profile of ALP genes from the brains and appendices of PD patients, showing that abnormalities in these genes can also occur on an epigenetic level. In particular, hypermethylation of several genes, such as ULK1 and HSC70, was observed [80].

Post-mortem studies on DLB patients demonstrated that LC3 co-localizes with LBs in both the hippocampus and the temporal cortex [81,82]. Moreover, ultrastructural analyses showed the presence of abundant and abnormal autophagosomes [82]. In DLB brains, levels of mechanistic mTOR were increased and ATG7 levels were decreased. Interestingly, such alterations may be responsible for a defective initiation of the macroautophagy pathway, resulting in an impaired fusion of lysosomes and, consequently, accumulation of α-syn aggregates. Nonetheless, this might result in the formation of enlarged autophagic vacuole-like structures that have been previously observed by other groups as well (Table 1) [82].

The connection between PD and autophagy is further strengthened by several findings coming from in vivo and in vitro studies, which also help to shed light on the pathways involved in the onset and advancement of these disorders.

### 2.2. Analyses of Autophagy in PD Patient-Derived Induced Pluripotent Stem Cells

The use of induced pluripotent stem cells (iPSCs) enables the generation of human-specific models to study neuronal degeneration when PD-relevant genes are mutated [87,88,89,90,91,92]. The generation of midbrain dopaminergic neurons using iPSCs, an important step for studying neurodegenerative diseases, was accomplished following a protocol using bone morphogenetic protein (BMP)/SMAD inhibition and activation [93,94]. Notably, signs of autophagic impairment were observed in two independent studies in neuronal cells derived from iPSCs characterized by triplication of the SNCA locus [88,92]. Supplementary observations of a link between autophagy and PD come from studies involving the most frequent LRRK2 mutated variant, LRRK2-G2019S, which is subject to enhanced LRRK2 kinase activity. A novel study combined the use of iPS-derived neurons carrying the LRRK2-G2019S mutation and knock-in transgenic mice demonstrating that, in the presence of mutated LRRK2, the transport of autophagic vesicles along the axons is disrupted, mainly owing to hyper-phosphorylation of the Rab29 GTPase, an important regulator of trafficking [95].

### 2.3. Analyses of Autophagy in Mouse Models of PD and DLB

A mouse model carrying the α-syn A53T mutation has been described to show neuronal accumulation of α-syn and severe motor dysfunctions [96,97]. Interestingly, transgenic mice overexpressing mutant α-syn additionally exhibited an induced macroautophagic activity with elevated Beclin-1 and LC3-II levels [98]. Conditional knock-in mice expressing A53T in dopaminergic neurons only presented mutant α-syn localized on the mitochondrial membrane, thus leading to reduced activity of the mitochondrial complex I and raised levels of mitochondrial autophagy (mitophagy) [99]. In both a transgenic mouse model and a neuronal cell line, it was reported that co-overexpression of α-syn and Beclin-1 could ameliorate the phenotype and lead to a reduction in α-syn accumulation, suggesting that modulation of macroautophagy could partially compensate the CMA deregulation induced by α-syn [100].

In another study, the authors showed that mice lacking ATP13A2 develop motor symptoms and endolysosomal abnormalities, but no alterations in the accumulation and aggregation of α-syn are evident [101]. A few reports focused on the involvement of VPS35, a key player in vesicular trafficking, in the development of PD. Transgenic mice lacking VPS35 display α-syn accumulation and neuronal loss in the SN, occurring together with increased Lamp-2A degradation, reinforcing the suggestion that upstream autophagic proteins are also key regulators of α-syn aggregation and LB formation [102]. A LRRK2-G2019S knock-in mouse model and primary cells carrying this mutation displayed expanded lysosomes with decreased degradative activity and concomitant upregulation of ATP13A2 levels [103]. It was recently observed that the same mouse model is characterized by altered mitophagy, an impairment that is strongly correlated with familial PD in patients [104]. LRRK2 can also interact with p62, and it has been shown that p62 is essential for the turnover of LRRK2 mediated by autophagy [105].

A knockout model of DJ-1 in mice and knockdown in SH-SY5Y cells presented elevated α-syn aggregation and inhibition of the CMA machinery. This effect was mediated by DJ-1 selectively accelerating degradation of Lamp-2A in the lysosomes, hence destabilizing the autophagic pathway [67]. Furthermore, a recent study investigated the effect of overexpression of DJ-1 in a PD rotenone-induced mouse model. The authors showed that DJ-1 can act as a neuroprotective factor when its expression is increased in astrocytes, generating a reduction in the levels of α-syn and increased levels of Lamp-2A, thus suggesting a non-cell-autonomous function and potential therapeutic target for astrocytes in PD [106]. It has also been reported that microglial cells can act as a neuroprotector and mediate the degradation of α-syn through the so-called synucleinphagy, a selective autophagic mechanism. In particular, synucleinphagy requires an upregulation of p62 levels to function, and impairment of microglial autophagy in a mouse model overexpressing α-syn resulted in a greater accumulation of aggregates and increased death of dopaminergic neurons [107,108]. For more details on the interplay of neurons, microglia, and astrocytes in the context of PD, see the recent review of MacMahon Copas et al. [109].

It was recently reported that knock-in Gba^L444P/WT^ transgenic mice also present, together with α-syn accumulation, reduced mitochondrial dynamics and impaired mitophagy [110]. Additionally, accumulation of α-syn was observed in a mouse model lacking another lysosomal enzyme, CatD. Remarkably, the same study reported that overexpression of CatD was able to rescue the phenotype by reducing α-syn aggregation in H4 neuroglioma cells, SH-SY5Y, and *C. elegans* [111].

Mutations in autophagy-related genes were also shown to play a role in α-syn accumulation in neuronal cells of various mouse models [112,113,114,115,116]. Transgenic Atg7 deficient mice showed autophagic impairment and progressive loss of neurons in the SN and striatum, along with increased α-syn and LRRK2 levels at the pre-synaptic compartment [114]. Moreover, knockout mice showed axonal dystrophy, highlighting a function of Atg7 in the trafficking of autophagic vesicles in the axons [112]. In the context of DLB, it has been proposed that a reduction in Atg7 and rise in mTOR levels could participate in the early phases of DLB; however, the detailed mechanisms remain to be elucidated [52].

### 2.4. Analyses of Autophagy in Different Cellular Models of PD and DLB

The first in vitro evidence of a connection between PD and autophagy came from a rat pheochromocytoma (PC12) cell line expressing either a mutant version of α-syn, A53T, or its wild type form. PC12 cells expressing mutant α-syn displayed an altered morphology and accumulation of lysosomal vacuolar structures, accompanied by observations of reduced lysosomal functions [117]. Other studies using the same cell line reported that wild type α-syn could be degraded through autophagy [28], especially through CMA, whereas the mutant form A53T failed to be degraded by CMA despite its high affinity for Lamp-2A. Thus, this resulted in an obstruction of the uptake and degradation of other substrates and triggered the activation of macroautophagy as a compensatory mechanism [118]. A similar phenotype was also observed using a human neuronal cell line [119]. Similarly to the mouse model described above, overexpressing Beclin-1 in PC12 cells carrying either the A53T or the A30P α-syn mutation resulted in a rescue of the phenotype and reduced α-syn accumulation [120]. In a recent study, midbrain dopaminergic neurons transfected with the A30P α-syn mutation exhibited a decreased autophagic activity shown by significantly elevated levels of p62 [30]. CMA can also be selectively blocked in the presence of α-syn post-translational modifications, as shown in the case of dopamine-modified α-syn in SH-SY5Y and mouse primary cells [121].

Yang Q. et al. investigated the role of myocyte enhancer factor 2D (MEF2D), a transcription factor associated with neuronal survival, in relation to autophagy and PD. It was observed that MEF2D can be degraded by CMA and, therefore, its levels were increased in a transgenic mouse model overexpressing α-syn, similar to samples derived from PD brains, thus indicating a direct connection between CMA regulation and neuronal survival [122]. As described before, altered levels of ATP13A2 were also reported in post-mortem PD samples, hence raising questions about the link between ATP13A2 and α-syn. In vitro knockdown of ATP13A2 in primary cortical neurons resulted in reduced lysosomal activity and consequent accumulation of α-syn, leading to elevated neuronal toxicity [123].

Along with LRRK2, DJ-1/PARK7 was also shown to be degraded by CMA in HEK293 cells and in a mouse dopaminergic progenitor SN4741 cell line [124]. Notably, studies reported a connection between DJ-1 and mitochondria. Knockdown of DJ-1 in M17 human dopaminergic neuroblastoma cells resulted in oxidative stress and compromised mitochondrial dynamics together with an increase in LC3-II levels, thus suggesting that DJ-1 might play a role in the modulation of autophagy in response to the presence of reactive oxygen species (ROS) [125].

As described before, GCase1 is a lysosomal enzyme acting downstream in the autophagic pathway. Nonetheless, it was observed that mutations in GBA1 in a neuroblastoma cell line are sufficient to induce an accumulation of α-syn and impaired lysosomal functions [126]. Analogous results were obtained in primary cells lentivirally transduced with a short hairpin RNA (shRNA) for knocking down GCase1 [127].

Taken together, in the past two decades, a multitude of studies are indicating that both macroautophagy and CMA participate in the onset and progression of PD and DLB. Impairment of the autophagic machinery was first observed in post-mortem samples from patients manifesting with an accumulation of vesicular structures and altered levels of various autophagy markers. The molecular and biological functions of such proteins have been extensively studied in vivo and in vitro. Remarkably, it seems that the accumulation of α-syn and the defective autophagic machinery are involved in a self-amplifying loop that drives the progression and worsening of the disease. In particular, accumulating evidence suggests that CMA has a higher impact on neuronal α-syn aggregation mechanisms compared with macroautophagy (Figure 2). However, many of the mechanisms regulating the interplay between autophagy-related proteins and PD-relevant genes remain to be elucidated. Therefore, autophagy not only represents an important aspect to study new pathogenic mechanisms of PD, but also offers the possibility of investigating autophagy modulation as a novel therapeutic target for neurodegenerative disorders.

## 3. The Role of Autophagy in Oligodendroglial ASPs

MSA is an adult-onset, highly progressive, rare, but fatal neurodegenerative disease with unknown etiology. Clinically, MSA is defined by levodopa-unresponsive Parkinsonism, cerebellar ataxia, pyramidal signs, and autonomic failure in various combinations [128]. To date, no curative treatment for MSA is available; only symptomatic alleviation is possible regarding the current treatment options. Stopping the progression of MSA is still an elaborate task, as the mechanisms of disease progression are far from elucidated.

MSA is suggested to be a primary oligodendrogliopathy, as α-syn accumulates essentially in the cytoplasm of oligodendroglial cells, creating so-called GCIs. The oligodendroglial injury caused by these aggregates may be a leading cause of neurodegeneration, as the damaged oligodendroglial cells may no longer properly support neurons [129]. The development of α-syn-positive GCIs is far from elucidated so far; thus, the focus of research lies on the mechanisms behind it. Different hypotheses on the emergence of α-syn and these deposits in oligodendroglia exist. One possible explanation is the cumulative expression of α-syn in oligodendroglial cells and, later, the accumulation of the protein in the cytoplasm. However, the expression profile of SNCA in oligodendroglia is reported differently in the literature. On one hand, reports exist claiming that SNCA mRNA, and thus the protein, is present in oligodendroglia [130,131]. On the other hand, some studies could not observe oligodendroglial SNCA mRNA in MSA brains or only a transient expression of α-syn during oligodendroglial maturation [132,133]. The incorporation of extracellular neuronal-derived or recombinant α-syn by oligodendroglial cells has been reported several times in various in vitro and in vivo studies, also suggesting several mechanisms of uptake including endocytosis via clathrin or exosomes [131,134,135,136,137,138,139,140,141,142,143]. Prion-like transmission is another proposed mechanism that might lead to the accumulation of toxic α-syn in oligodendroglial cells, suggesting that α-syn behaves like a prion, thereby inducing MSA. However, in no study did significant amounts of α-syn-positive oligodendroglial inclusions occur; mostly, neuronal aggregates developed when adding α-syn from MSA brains to α-syn transgenic mice. Interestingly, prion-like behaviour of MSA brain homogenates could only be detected in mice having an A53T mutation, which is an α-syn mutation that occurs in PD patients and does not explain how GCIs and MSA develop driven by a prion-like manner [144,145,146]. Furthermore, α-syn seems to lack other different properties of a real prion protein, among others, the human transmission of α-syn, suggesting that MSA is not a real “prionopathy” [147,148].

### 3.1. Autophagy Involvement in Human Pathology of MSA

As discussed already in the introduction, in the cell, misfolded α-syn is degraded preferentially via the autophagy machinery [31,86,149]. If the macroautophagic machinery is not working appropriately, it can result in the aggregation of α-syn and, furthermore, in the formation of GCIs in oligodendroglial cells. In MSA brain tissue, numerous GCIs were found to be immunoreactive for LC3, a marker of macroautophagy. Thus, these GCIs were not positively stained for γ-aminobutyric-acid type A receptor-associated proteins (GABARAPs), suggesting a reduced maturation of autophagosomes [34]. In another study using MSA brain tissue, GCIs were stained positively for LC3, as well as for the ubiquitin-binding protein p62 and ubiquitin, confirming an upregulation of macroautophagy in oligodendroglial cells in MSA [149]. In a case report, p62-positive GCIs were present in MSA brain tissue of a patient with a prolonged clinical course [150]. Another case report claimed an increased amount of Beclin-1 corresponding to the area of oligodendroglial cell loss [84]. Numerous GCIs were positively stained for HSC70 in MSA compared with healthy brains in a different study by Kawamoto and colleagues [85]. The integration of autophagic proteins in GCIs indicates that the cell is starting an autophagy pathway and trying to clear the increased amount of α-syn, which is a cytoprotective mechanism [151]. Yet, the cell is not able to cope with the large amounts of toxic protein, leading to a reduction in autophagy proteins involved in downstream autophagic events, suggesting a defective macroautophagy maturation or suppressing CMA.

In a recent study, Miki Y. and colleagues proposed that upstream autophagy proteins also play a role in MSA pathogenesis. They could show that the autophagy/beclin1 regulator 1 (AMBRA1), an upstream protein of macroautophagy, is present in GCIs in brains of MSA patients, and that AMBRA1 silencing in neuronal cells leads to α-syn aggregate formation. It is suggested that AMBRA1 is connected to the recognition of abnormal and phosphorylated α-syn, hence activating autophagy to degrade these inclusions [83]. It is possible that the accumulation of AMBRA1 in GCIs might occur, trying to get rid of the large amount of α-syn by the cell, yet exceeding the cellular efficiency. Another mechanism involved in regulating macroautophagy has been proposed in another study, describing an increase in microRNA-101 (miR-101), among others, which targets different autophagy-related genes including *RAB5A* as well as *MTOR*, combined with a downregulation of the target gene *RAB5A* in the striatum of MSA patients [152]. Overexpression of miR-101 led to deficits in the autophagy machinery and, therefore, to accumulation of α-syn in oligodendroglial cells, substantiating the role for autophagy involvement in aggregation formation in MSA. The authors suggest that a dysregulation of miRNAs leads to a decreased expression of autophagy proteins, augmenting the accumulation of abnormal α-syn [152].

### 3.2. Analyses of Autophagy in Human-Derived Induced Pluripotent Stem Cells

The application of iPSCs was utilized to generate a human MSA iPSC model using human fibroblasts [132,153,154]. In the human MSA iPSC model, the authors could show a higher amount of the autophagic protein LC3-II in MSA patient-derived compared with control iPSC-derived neuronal cells. Furthermore, a decrease in the LC3-II flux was found in MSA-patient-derived iPSC, suggesting a disrupted autophagic flow, yet only in neuronal cells [154]. Even though these results were only shown in neuronal cells, this could be an indication for an overall disrupted autophagic machinery in MSA.

### 3.3. Analyses of Autophagy in Mouse Models of MSA

Using a transgenic MSA mouse model with overexpression of human α-syn under the oligodendroglial proteolipid protein promoter (PLP) [155], the levels of the macroautophagy protein LC3 were again increased when compared with wild type mice [156]. In another MSA mouse model, overexpressing α-syn under the myelin basic protein (MBP) promoter, an anti-miR-101 construct was delivered lentivirally into the striatum, enhancing macroautophagy, thereby decreasing the accumulation of α-syn in oligodendroglial cells [152]. The outcomes of these animal studies also suggest an involvement of macroautophagy in the aggregation mechanism of α-syn in oligodendroglial cells in MSA.

### 3.4. Analyses of Autophagy in Different Cellular Models of MSA

Different cell culture studies using oligodendroglial cells could also indicate an involvement of macroautophagy in aggregation formation of α-syn. Proteasomal inhibition in rat brain oligodendroglial cells led to the aggregation of α-syn and to the recruitment of LC3 and p62 to these inclusions, implying a close connection between the proteasomal system and the macroautophagy machinery [149]. In vitro induced mitochondrial impairment, oxidative stress, or macroautophagy inhibition caused the aggregation of exogenously added α-syn in rat brain oligodendroglial cells and/or in an oligodendroglial cell line [139]. In a recent study, it was found that α-syn preformed fibrils can facilitate the increase of endogenous α-syn in oligodendroglial precursor cells (OPCs) owing to a deficient autophagy machinery. The macroautophagy deficiency was confirmed by showing an increase in p62 and the LC3-II/LC3-I ratio in these OPCs [131]. Yet, in another study, the pharmacological inhibition of macroautophagy, as well as the knockdown of LC3 in an oligodendroglial cell line, only had limited effects on the accumulation formation of α-syn [136]. However, this outcome might correlate with the limited incubation time of oligodendroglial cells to extracellular α-syn species, because, in a different study, the uptake of α-syn from the extracellular space was suggested to be time-dependent [142]. These data suggest that, possibly, different forms of stress, including proteasomal, mitochondrial, oxidative, and autophagic stress, might be interconnected and must occur to create the full-blown pathology in MSA.

Summarizing, the afore-mentioned studies strengthen the assumption that an involvement of the autophagic pathway takes place in the initiation and progression of α-syn inclusion formation and, therefore, in disease progression in MSA. Although robust data on CMA regarding the recognition and degradation of α-syn in neuronal cells exist [29,118,121], only one human post-mortem study was published on an involvement of CMA in oligodendroglial inclusion formation in MSA so far, showing GCIs positively immunostained for HSC70 [85]. The available results introduce the speculation that HSC70 tries to bind and escort α-syn to the lysosome in oligodendroglial cells in MSA; however, it is not able to do so, thereby blocking CMA in its early stages, thus macroautophagy might be the more active degradation pathway in oligodendroglia (Figure 2). Yet, further studies will be needed, especially on CMA, to deepen and substantiate the current knowledge on the role of autophagy in GCI formation in oligodendroglial cells in MSA.

## 4. Concluding Remarks

To date, the involvement of autophagy in neurodegenerative diseases is well-acknowledged. As the pathological hallmark of most neurodegenerative diseases is the accumulation of different proteins in neuronal or glial cells, an impairment in the degradation process of these proteins is highly likely. Especially, in ASPs, the aggregation of α-syn and other proteins leads to the pathological hallmarks, which are LBs in PD and DLB and GCIs in MSA, initiating and/or progressing disease advancement. Macroautophagy and CMA have been shown to be major mechanisms involved in the attempt to reduce misfolded and aggregated α-syn in neuronal cells. In PD and DLB, evidence on the disturbances of macroautophagy and CMA was found in multiple studies. In MSA, a proven involvement of CMA in the inclusion formation of α-syn is still missing, yet different studies suggest a contribution of macroautophagy in disease initiation and progression. However, it is possible that, in oligodendroglial cells, CMA is repressed early when accumulated α-syn needs to be degraded. One possible mechanism could be that CMA is dependent on the form of the aggregated α-syn. Various toxic oligomeric α-syn variants were detected in PD brains [157,158] and different α-syn strains were described to result in different ASPs [159]. It could be that these different variants might also induce different pathways of degradation in PD, DLB and MSA, as well as in neurons and oligodendroglial cells. However, more studies will be needed to elucidate an overall picture on the role of macroautophagy and CMA in ASPs. If defined autophagic mechanisms or dysfunctions can be described, these studies will provide ideal candidates for drug development and, furthermore, might be involved in halting the progression of these neurodegenerative diseases.

## Figures and Tables

**Figure 1 cells-10-03143-f001:**
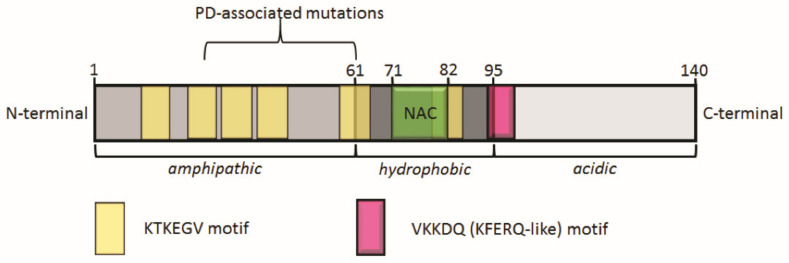
Schematic representation of full-length α-synuclein structure (140 amino acids). α-synuclein consists of an amphipathic region from amino acid 1 to 61 with several repetitions of the conserved KTKEGV motif. This region is involved in membrane binding. The hydrophobic part of α-synuclein extends from amino acid 62 to 95 and comprises the non-amyloid β-component (NAC) region, which is important for promoting aggregation. From amino acids 96 to 140, α-synuclein features an acidic region, which is involved in ligand binding. Moreover, α-synuclein holds a VKKDQ (KFERQ-like) motif that is the binding site for heat shock cognate 70 (Hsc70) inducing degradation via chaperone-mediated autophagy (CMA) [25].

**Figure 2 cells-10-03143-f002:**
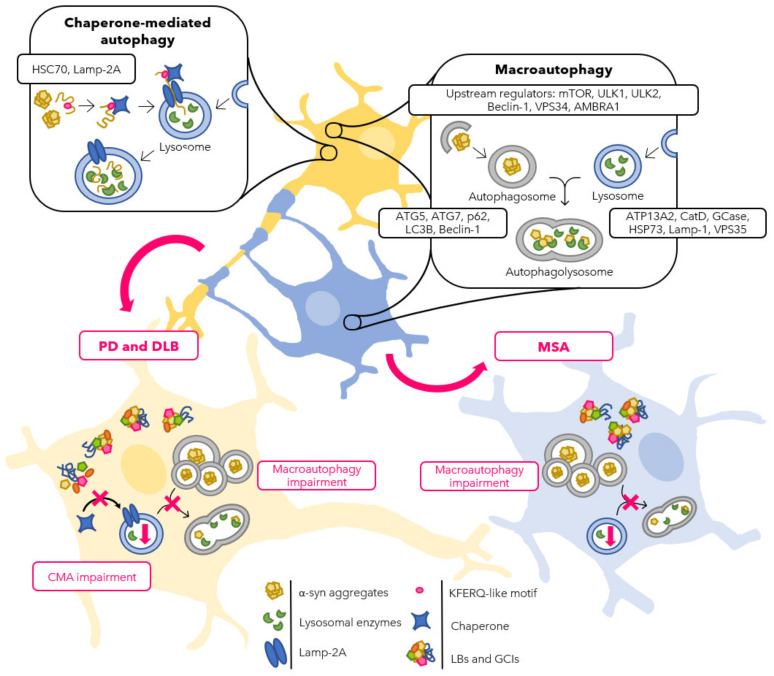
Schematic representation of the autophagic flux in neurons (yellow) and oligodendrocytes (blue) in normal and PD, DLB, and MSA conditions. Physiologically, both CMA and macroautophagy pathways are involved in protein degradation in the cytoplasm of neurons, whereas macroautophagy seems to be the more effective mechanism in oligodendrocytes. It has been reported that, in the context of ASPs, an impairment of the autophagic machinery is involved. PD and DLB neurons (bottom left) present with LBs in the cytoplasm accumulating α-syn together with other proteins that can be degraded by neither macroautophagy nor CMA, such as LRRK2, DJ-1, and UCHL1. Moreover, other autophagy-related proteins, such as AMBRA1, p62, and LC3B, accumulate in LBs. Eventually, macroautophagy impairment results in an accumulation of autophagosomes and reduction in lysosomal activity. MSA oligodendrocytes (bottom right) show defects as well in the macroautophagic pathway with accumulation of GCIs containing α-syn, AMBRA1, p62, LC3B, and HSC70, together with an increased number of autophagosomes and diminished degradation in the lysosomes.

**Table 1 cells-10-03143-t001:** Evidence of autophagy involvement found in post-mortem brain tissue of PD, DLB, and MSA patients.

AutophagyMarkers	Function	PD	DLB	MSA
AMBRA1	upstream regulator	increased immunoreactivity [61]	increased immunoreactivity [61]	increased protein levels and immunoreactivity [83]
ATG7	upstream regulator	-	decreased protein levels and immunoreactivity [82]	-
ATP13A2	lysosomal ATPase	decreased immunoreactivity [62]	-	-
Beclin-1	autophagosome generation	increased protein levels [61]	partially increased protein levels [61]	increased immuno-reactivity [84]
Cathepsin D	lysosomal hydrolase	decreased immunoreactivity [72]	increased immunoreactivity [82]	-
HSC70	chaperone involved in CMA	decreased protein levels [59]	-	increased immuno-reactivity [85]
Lamp-1	lysosomal membrane glycoprotein	decreased immunoreactivity [72]	-	-
Lamp-2A	membrane receptor for CMA	decreased protein levels, immunoreactivity in few LBs [59,65]	-	-
LC3B	autophagosome generation	increased immunoreactivity [58], increased LC3B-II protein levels [59]	increased immunoreactivity [81,82]	increased immuno-reactivity [34,86]
mTOR	upstream regulator	-	increased protein levels and immunoreactivity [82]	-
p62	upstream regulator	increased immunoreactivity [60]	-	increased immuno-reactivity [86]
ULK1/2	upstream regulator	increased immunoreactivity [61]	increased immunoreactivity [61]	-
VPS34	vesicle trafficking	increased immunoreactivity [61]	increased protein levels and immunoreactivity [61]	-

## Data Availability

No new data were created or analyzed in this study. Data sharing is not applicable to this article.

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
