# Peer review of "Autophagy in α-Synucleinopathies—An Overstrained System"

_cells, 2021, doi:10.3390/cells10113143_

Round 1

Reviewer 1 Report

The manuscript entitled “Autophagy in a-synulceinopathies – an overstrained system” discusses the hypothesis that autophagy-lysosomal dysfunction underlies alpha-synuclein Lewy pathology. The review/discussion is well timed considering several recent –omics reports that strongly implicate the autophagy-lysosomal pathway in the disease process. The authors do a good job at summarizing some of the literature in this area. The manuscript could benefit from a stronger stance on autophagy-lysosome role in synucleinopathies, for example discussion about how autophagy dysfunction might unify several clinically distinct diseases (i.e. MSA, PD, and DLB).  Correspondingly, taking

Specific comments below:

-Line 15. Aggregates occur in distinct cellular compartments (terminals, dendrites, axons, cell body). Lewy bodies refer to aggregates observed in the cell body. Lewy bodies are simple to detect, but advent of excellent synuclein antibodies makes it clear that the pathology is distributed in the entire cell.

-Line 22. Chicken or egg. Autophagy activation increases asyn, or does increased asyn activate autophagy? Could be both ways. Authors could highlight this point.

-Line 43. Remove “profound”

-Table 1 might be more useful with more information such as: gene function and assay that was used for the conclusions was reach.

-Figure 1: Amphipatic should be “amphipathic”

-In concluding remarks: “Uncontested” is probably too strong wording.

-The author assumes that neuronal Lewy pathology is homogenous. That is unknown, but LP may result from the same pathogenic process across synucleinopathies. Author might want to state this assumption, or bring attention to this assumption.

-Line 110: “invited to read” might be better to say for detailed review see XXX.

Line 133. A-syn accumulation as a key hallmark of these diseases, but it remains to be determined if the accumulation is driving the disease process.

-Line 154. The statement sound that DLBA and PD are clearly distinct entities. DLB and PD a-syn pathology distribution are very similar. Generally, DLB has more prominent cortex pathology. However many DLB cases do have marked midbrain Lewy pathology (LB and LN) and many PD cases have marked cortex pathology. The distinction between the two diseases is largely driven by clinical diagnosis (symptom onset). It might help the author’s arguments to highlight the commonalities between DLB and PD – and what evidence is there that they differ pathologically, if at all.

-Paragraph starting on line 186 – not clear that endogenous dimers exist. Might be more accurate to state that impairments of CMA machinery impair asyn turnover, increase levels, and this possibly promotes aggregation.

-Authors should cite and discuss Gordevicius J et al. 2021, which found epigenetic silencing of autophagy genes in both peripheral tissues and brain of individuals with PD.

-For MSA section, authors discuss the interesting finding that MSA pathology did not seed GCI’s in an animal model. Instead, only neuronal pathology was observed. Does this suggest autophagy dysfunction occurs in neurons or glia?

Author Response

The manuscript entitled “Autophagy in a-synulceinopathies – an overstrained system” discusses the hypothesis that autophagy-lysosomal dysfunction underlies alpha-synuclein Lewy pathology. The review/discussion is well timed considering several recent –omics reports that strongly implicate the autophagy-lysosomal pathway in the disease process. The authors do a good job at summarizing some of the literature in this area. The manuscript could benefit from a stronger stance on autophagy-lysosome role in synucleinopathies, for example discussion about how autophagy dysfunction might unify several clinically distinct diseases (i.e. MSA, PD, and DLB).  Correspondingly, taking

Response: We thank the reviewer for this comment. We also think that autophagy dysfunction might unify these clinically distinct diseases. We added a sentence about this in line 144: “Yet even though there are differences regarding the exact mechanisms behind α-syn aggregation, autophagy dysfunction seems to be involved in all three clinically distinct diseases, i.e., PD, DLB and MSA.

Specific comments below:

-Line 15. Aggregates occur in distinct cellular compartments (terminals, dendrites, axons, cell body). Lewy bodies refer to aggregates observed in the cell body. Lewy bodies are simple to detect, but advent of excellent synuclein antibodies makes it clear that the pathology is distributed in the entire cell.

Response: We thank the reviewer for the valuable comment. We changed that in the abstract accordingly (see line 15). Thus, it now reads as follows “The aggregation of α-synuclein in the cell body of neurons, giving rise to the so-called Lewy bodies (LBs), is the major characteristic for PD and DLB, whereas the accumulation of α-synuclein in oligodendroglial cells, so-called glial cytoplasmic inclusions (GCIs), is the hallmark for MSA.”

-Line 22. Chicken or egg. Autophagy activation increases asyn, or does increased asyn activate autophagy? Could be both ways. Authors could highlight this point.

Response: We appreciate the comment by the reviewer. Thus, we have included a sentence in the introduction (line 23) stating the following: “To date, it is not entirely clear what the trigger point in disease initiation is: whether autophagy dysfunction alone suffices to increase α-synuclein or whether α-synuclein is the pathogenic driver.”

-Line 43. Remove “profound”

Response: We appreciate the reviewer’s comment and removed the word profound in line 47.

-Table 1 might be more useful with more information such as: gene function and assay that was used for the conclusions was reach.

Response: We thank the reviewer for this valuable suggestion. We added the protein function and included what was checked (protein levels, immunoreactivity) in the table.

-Figure 1: Amphipatic should be “amphipathic”

Response: We thank the reviewer for pointing out this typo. We corrected the mistake in Figure 1, accordingly.

-In concluding remarks: “Uncontested” is probably too strong wording.

Response: We agree with the reviewer and changed uncontested to “well-acknowledged” (line 521).

-The author assumes that neuronal Lewy pathology is homogenous. That is unknown, but LP may result from the same pathogenic process across synucleinopathies. Author might want to state this assumption, or bring attention to this assumption.

Response: We thank the reviewer for this comment. We added a sentence in lines 171-174: “To date, there is no study reporting that PD and DLB share the same pathogenic process, nonetheless, many common features have been observed in patients in the context of autophagy, as will be described below.

-Line 110: “invited to read” might be better to say for detailed review see XXX.

Response: We agree with the reviewer and changed it in line 115 as suggested. Now it reads as follows: “For a detailed report on macroautophagy and its key players, see the reviews from Dikic I. & Elazar Z. and Yang Z. & Klionsky D.J. [42,43].”

Line 133. A-syn accumulation as a key hallmark of these diseases, but it remains to be determined if the accumulation is driving the disease process.

Response: We thank the reviewer for the comment, and we removed that accumulation is a driving disease process (line 137).

-Line 154. The statement sound that DLBA and PD are clearly distinct entities. DLB and PD a-syn pathology distribution are very similar. Generally, DLB has more prominent cortex pathology. However many DLB cases do have marked midbrain Lewy pathology (LB and LN) and many PD cases have marked cortex pathology. The distinction between the two diseases is largely driven by clinical diagnosis (symptom onset). It might help the author’s arguments to highlight the commonalities between DLB and PD – and what evidence is there that they differ pathologically, if at all.

Response: We appreciate this comment by the reviewer. We have improved accordingly this section starting from line 160 as follows: “DLB is mainly characterized by the presence of diffuse LBs in the SN, the cerebral cortex and the subcortical nuclei, with a more marked cortex pathology compared to PD [51]. However, PD and DLB share many similarities in LBs occurrence and localization and can be distinguished mainly thanks to diagnostic criteria. In particular, clinical manifestations of DLB include cognitive deficits associated with a parkinsonian phenotype, and fluctuations in attention and awareness, often accompanied by visual hallucinations [52,53].

-Paragraph starting on line 186 – not clear that endogenous dimers exist. Might be more accurate to state that impairments of CMA machinery impair asyn turnover, increase levels, and this possibly promotes aggregation.

Response: We agree with the reviewer that it is not clear whether endogenous dimers exist. However, we do not write about endogenous dimers here, but we write about dimers that build in the cell due to some defect in the autophagic machinery.

-Authors should cite and discuss Gordevicius J et al. 2021, which found epigenetic silencing of autophagy genes in both peripheral tissues and brain of individuals with PD.

Response: We appreciate the reviewer’s suggestion to include this novel study. We added it beginning from line 231: “Interestingly, a recent study has investigated the methylation profile of ALP genes from brains and appendix of PD patients, showing that abnormalities in these genes can also occur on an epigenetic level. In particular, hypermethylation of several genes, such as ULK1 and HSC70, was observed.

-For MSA section, authors discuss the interesting finding that MSA pathology did not seed GCI’s in an animal model. Instead, only neuronal pathology was observed. Does this suggest autophagy dysfunction occurs in neurons or glia?

Response: We thank the reviewer for this question. Actually, these studies only checked on the prion properties of brain homogenates of MSA brains and did not check on autophagic pathways as far as we know. Interestingly, they found this synuclein propagation only in mice homozygous or heterozygous for a mutant A53T human α-synuclein transgene, which is a common mutation associated with PD. This could of course induce the neuronal accumulations, as the neuronal cells might have been already susceptible due to the mutation. In synuclein wild type mice, no propagation could be detected. Regarding the development of MSA, these studies do not explain how these MSA prions should induce oligodendroglial accumulation in MSA, as A53T is not a mutation that occurs in MSA patients at all. To bring more clarity to this, we added another sentence to this part and discussed that the aggregates only occurred in A53T mutated mice (lines 420-425). It changed as follows: “However, in no study significant amounts of α-syn-positive oligodendroglial inclusions occurred, mostly neuronal aggregates developed when adding α-syn from MSA brains to α-syn transgenic mice. Interestingly, prion-like behaviour of MSA brain homogenates could only be detected in mice having an A53T mutation, which is an α-syn mutation that occurs in PD patients and does not explain how GCIs and MSA develop driven by a prion-like manner“. Yet, in our opinion, any statement from us on the role of autophagy in these models would be far-fetched, as there are no indications of the involvement of a dysfunctional autophagy pathway here. Further studies would be needed to check on autophagy mechanisms in such a prion-propagation model.

Reviewer 2 Report

On the whole, this is a nice review. The abstract and introduction are well-written, and the review is well-referenced. I have specific suggestions below, mostly related to dividing section 2 and 3 so as to make it more readable.

Major

  1. Section 2 and section 3 would be greatly improved by sub-headings and ensuring that each paragraph has a main idea. They could be organized either by model system (patient samples, PC12, iPSC, mice, etc) or by PD mechanism (A53T, over-expression, LRRK2, etc) or by autophagy protein (beclin, Lamp2, etc), but some organization is required. For example, on page 6, the first paragraph starts being about PC12 cells and beclin-1, then moves to mice, then moves to an explanation of how iPSC are made. This is a lot for one paragraph and makes the paper difficult to read. The next paragraph is another example of this – it starts being about MEF2D and then is suddenly about ATP13A2, then VPS35. I think sub-headings would help clarify the main topic for each paragraph.

Minor

  1. Line 32 – PD and DLB also feature inclusions in the PNS
  2. Lines 79-81, please add references for this claim (I agree, but references should be cited)
  3. Lines 92-94, grammar is wrong
  4. Line 99, should this read “ATGs, AuTophaGy” so that that the word matches the acronym?
  5. Lines 155-156 – I would not describe DLB as having “psychiatric fluctuations” - it is really a fluctuation in awareness. Please see for example McKeith et al Neurology 2017 for diagnostic criteria for DLB.  
  6. Lines 414-418 – the authors assert that integration of autophagic proteins indicates a cytoprotective mechanism of the cell – Is this really known? At a minimum please add references for this claim.

Author Response

On the whole, this is a nice review. The abstract and introduction are well-written, and the review is well-referenced. I have specific suggestions below, mostly related to dividing section 2 and 3 so as to make it more readable.

Major

  1. Section 2 and section 3 would be greatly improved by sub-headings and ensuring that each paragraph has a main idea. They could be organized either by model system (patient samples, PC12, iPSC, mice, etc) or by PD mechanism (A53T, over-expression, LRRK2, etc) or by autophagy protein (beclin, Lamp2, etc), but some organization is required. For example, on page 6, the first paragraph starts being about PC12 cells and beclin-1, then moves to mice, then moves to an explanation of how iPSC are made. This is a lot for one paragraph and makes the paper difficult to read. The next paragraph is another example of this – it starts being about MEF2D and then is suddenly about ATP13A2, then VPS35. I think sub-headings would help clarify the main topic for each paragraph.

Response: We thank the reviewer for this comment and agree that it makes sense to add some subheadings. We divided the PD/DLB and MSA parts now and added subheadings that divide the chapters by model systems, as suggested. Added subsections include human pathology, autophagy in human-derived pluripotent stem cells, autophagy in animal models and autophagy in cellular models.

Minor

  1. Line 32 – PD and DLB also feature inclusions in the PNS

Response: We thank the reviewer for this comment. We added the inclusion in the PNS in line 34. Now it reads as follows: “Inclusions of α-syn and neuroinflammatory response have also been reported to occur in the peripheral nervous system (PNS) [1–3].”

  1. Lines 79-81, please add references for this claim (I agree, but references should be cited)

Response: We agree with this comment and added the appropriate references: Xilouri M. 2016, Movement Disorders.

  1. Lines 92-94, grammar is wrong

Response: We thank the reviewer for pointing out this mistake. We changed the sentence as follows (line 96-98): “Macroautophagy (commonly called autophagy) begins with the generation of autophagosomes, which are double-lipid membrane structures that are involved in degrading cytosolic constituents, such as organelles and proteins.”

  1. Line 99, should this read “ATGs, AuTophaGy” so that that the word matches the acronym?

Response: We appreciate the comment of the reviewer and changed it to AuTophaGy so that it matches the acronym (line 103).

  1. Lines 155-156 – I would not describe DLB as having “psychiatric fluctuations” - it is really a fluctuation in awareness. Please see for example McKeith et al Neurology 2017 for diagnostic criteria for DLB.

Response: We appreciate this comment by the reviewer. We have tried to improve this section and added the suggested reference, starting from line 160 as follows: “DLB is mainly characterized by the presence of diffuse LBs in the SN, the cerebral cortex and the subcortical nuclei, with a more marked cortex pathology compared to PD [51]. However, PD and DLB share many similarities in LBs occurrence and localization and can be distinguished mainly thanks to diagnostic criteria. In particular, clinical manifestations of DLB include cognitive deficits associated with a parkinsonian phenotype, and fluctuations in attention and awareness, often accompanied by visual hallucinations [52,53].

  1. Lines 414-418 – the authors assert that integration of autophagic proteins indicates a cytoprotective mechanism of the cell – Is this really known? At a minimum please add references for this claim.

Response: We agree with the reviewer. We have changed the sentence, as it is true that the integration of autophagic proteins itself is not a cytoprotective mechanism, rather the aim of the cell to get rid of the accumulation through autophagy. We have also added a reference, a review discussing the cytoprotective role of autophagy. Line 444: “The integration of autophagic proteins in GCIs indicates that the cell is starting an autophagy pathway and trying to clear the increased amount of α-syn, which is a cytoprotective mechanism [153].

Round 2

Reviewer 2 Report

The authors have sufficiently addressed my comments. This is a nice review and a helpful addition to the field.